# The Effect of Synbiotic Supplementation on Uremic Toxins, Oxidative Stress, and Inflammation in Hemodialysis Patients—Results of an Uncontrolled Prospective Single-Arm Study

**DOI:** 10.3390/medicina59081383

**Published:** 2023-07-28

**Authors:** Teodor Kuskunov, Eduard Tilkiyan, Daniel Doykov, Krasimir Boyanov, Anelia Bivolarska, Bozhidar Hristov

**Affiliations:** 1Department of Propaedeutics of Internal Diseases, Medical Faculty, Medical University of Plovdiv, 4000 Plovdiv, Bulgaria; kuskunov@abv.bg; 2Hemodialysis Unit, University Hospital “Kaspela”, 4000 Plovdiv, Bulgaria; 3Second Department of Internal Diseases, Section “Nephrology”, Medical Faculty, Medical University of Plovdiv, 4000 Plovdiv, Bulgaria; eet64@yahoo.com; 4Nephrology Clinic, University Hospital “Kaspela”, 4000 Plovdiv, Bulgaria; 5Second Department of Internal Diseases, Section “Gastroenterology”, Medical Faculty, Medical University of Plovdiv, 4000 Plovdiv, Bulgaria; daniel_doykov@abv.bg; 6Gastroenterology Clinic, University Hospital “Kaspela”, 4000 Plovdiv, Bulgaria; 7Department of Medical Biochemistry, Faculty of Pharmacy, Medical University of Plovdiv, 4000 Plovdiv, Bulgaria; krasimir.boyanov@mu-plovdiv.bg (K.B.); anelia.bivolarska@mu-plovdiv.bg (A.B.)

**Keywords:** chronic kidney disease, hemodialysis, indoxyl sulfate, p-cresyl sulfate, interleukin-6, malondialdehyde, synbiotic

## Abstract

*Introduction*: Numerous studies to date have shown that the development of dysbiotic gut microbiota is a characteristic finding in chronic kidney disease (CKD). A number of uremic toxins progressively accumulate in the course of CKD, some of them generated by the intestinal microbiome, such as indoxyl sulfate (IS) and p-cresyl sulfate (p-CS). They are found to be involved in the pathogenesis of certain complications of uremic syndrome, including low-grade chronic inflammation and oxidative stress. The aim of the present study is to research the serum concentration of IS and p-CS in end stage renal disease (ESRD) patients undergoing conventional hemodialysis, as well as to study the possibilities of influencing some markers of inflammation and oxidative stress after taking a synbiotic. *Materials and Methods*: Thirty patients with end-stage renal disease (ESRD) undergoing hemodialysis treatment who were taking a synbiotic in the form of *Lactobacillus acidophilus* La-14 2 × 10^11^ (CFU)/g and prebiotic fructooligosaccharides were included in the study. Serum levels of total IS, total p-CS, Interleukin-6 (IL-6), and Malondialdehyde (MDA) were measured at baseline and after 8 weeks. *Results*. The baseline values of the four investigated indicators in the patients were significantly higher—p-CS (29.26 ± 58.32 pg/mL), IS (212.89 ± 208.59 ng/mL), IL-6 (13.84 ± 2.02 pg/mL), and MDA (1430.33 ± 583.42 pg/mL), compared to the results obtained after 8 weeks of intake, as we found a significant decrease in the parameters compared to the baseline—p-CS (6.40 ± 0.79 pg/mL, *p* = 0.041), IS (47.08 ± 3.24 ng/mL, *p* < 0.001), IL-6 (9.14 ± 1.67 pg/mL, *p* < 0.001), and MDA (1003.47 ± 518.37 pg/mL, *p* < 0.001). *Conclusions*: The current study found that the restoration of the intestinal microbiota in patients with CKD significantly decreases the level of certain uremic toxins. It is likely that this favorably affects certain aspects of CKD, such as persistent low-grade inflammation and oxidative stress.

## 1. Introduction

The human gut microbiota consists of over 100 trillion microbial cells, and its species diversity is unique to each individual and includes 500 to 1000 bacterial species [1,2]. Different parts of the gastrointestinal tract are colonized in varying quantitative proportions, with the predominant bacterial strains being *Bacteroidetes, Actinobacteria,* and *Frimicutes* [3,4]. In their study, The Human Microbiome Project, Gevers et al. [5] reveal the vital importance and impact of the gut microbiome on human health and disease development.

The progression of chronic kidney disease (CKD) to end-stage renal disease (ESRD) is associated with both a qualitative and a quantitative change in the composition of the intestinal microbiota, which leads to the development of a dysbiotic intestinal condition in these patients [6,7,8,9]. In 1996, Simenhoff et al. [6] reported that patients with uremia had a significantly higher number of aerobic and anaerobic micro-organisms inhabiting the duodenum and small intestine compared to healthy individuals. Hida et al. [7] demonstrated an increased number of aerobic bacteria from *the Enterobacteria* and *Enterococci families* combined with a decreased count of *Bifidobacterium species* in hemodialysis patients, as well as much higher *Clostridium perfringens* colonization in the same patients. Later, Vaziri et al. [8] reported a significantly higher abundance of *Brachybacterium, Catenibacterium, Halomonadaceae, Enterobacteriaceae, Moraxellaceae, Polyangiaceae, Thiothrix, Nesterenkonia*, and *Pseudomonadaceae* families in ESRD patients compared to healthy controls. A distinctive feature of some of these micro-organisms is a pronounced urease and uricase activity, as well as increased synthesis of indoles and phenols [9]. This, on its behalf, leads to an increased production of toxic substances and their subsequent accumulation in the blood circulation, due to reduced renal clearance in these patients [10,11]. Additionally, Wong et al. [9] indicated that the intestinal dysbiosis in CKD causes an impaired synthesis of useful metabolites such as short-chain fatty acids (SCFA), which are the main source of nutrition for intestinal bacterial proliferation.

To date, more than 150 molecules regarded as toxic in the context of uremia have been discovered [12]. Some of them are classified as protein-bound uremic toxins, including indoxyl sulfate and p-cresyl sulfate [13]. The aforementioned are products of the intestinal microbial metabolism of the aromatic amino acids—tyrosine, phenylalanine, and tryptophan—and accumulate progressively in the course of CKD [14,15,16]. Tryptophan is metabolized mainly into indole, while tyrosine and phenylalanine are metabolized into para-cresol, both of which are eventually absorbed and undergo partial detoxification in the liver mainly by sulfation to indoxyl sulfate (IS) and para-cresyl sulfate (p-CS), respectively [17].

In preserved renal function, the predominant mechanism of excretion of p-CS and IS in the urine is through tubular secretion [18,19]. In patients undergoing conventional hemodialysis treatment, however, their elimination is largely limited due to their high affinity to plasma proteins [20]. High serum concentrations of IS and p-CS in CKD have a variety of detrimental effects on the normal function of multiple organs and systems. Elevated serum IS levels are associated with numerous negative effects on the cardiovascular system including increased oxidative stress on the vascular endothelium, decreased vascular elasticity and vascular smooth muscle proliferation, aortic calcification, and increased cardiovascular-associated and total mortality in CKD patients [21,22,23]. Lin et al. [24] reported that IS and p-CS increase the risk of peripheral vascular disease and vascular access route thrombosis in hemodialysis patients. The role of IS in the pathophysiology of bone disorders in CKD is associated with the suppression of osteoclast activity and development of adynamic bone disease [25,26]. Nii-Kono et al. [27] hypothesize that IS stimulates oxidative stress in osteoblasts and promote the development of resistance to parathormone, leading to a reduced rate of bone formation. In another study of theirs, Lin et al. [28] establish a positive association between high serum levels of IS and those of fibroblast growth factor 23 (FGF-23). IS has been found to have a marked profibrotic effect on the myocardium, to stimulate myocardiocyte hypertrophy and to increase the risk of atrial fibrillation [29,30,31]. A correlation has been found between IS levels and the development of anemic syndrome in patients with CKD, as it inhibits erythropoiesis, suppresses the activity of erythropoietin, and potentiates eryptosis—the programmed cell death of erythrocytes [32,33,34]. Similar to IS, elevated serum levels of p-CS increase the risk of cardiovascular and total mortality in patients with CKD [35,36]. P-CS also stimulates profibrotic and inflammatory responses as well as oxidative stress [37,38,39].

## 2. Materials and Methods

### 2.1. Study Settings and Population

Thirty patients undergoing chronic hemodialysis treatment in the Hemodialysis Unit of University Hospital Kaspela, Plovdiv, Bulgaria were included in the study. The laboratory analysis of the samples was carried out in the Department of Medical Biochemistry of the Medical University of Plovdiv. The study was approved by the scientific ethics committee of Medical University of Plovdiv (protocol No. 1/19.01.2023) and all subjects gave written informed consent. Design: Prospective, quasi-experimental single-center study.

We tested the serum levels of indoxyl sulfate, p-cresyl sulfate, malondialdehyde, and interleukin-6 in 30 patients (women *n* = 13, men *n* = 17) undergoing conventional hemodialysis treatment at baseline and after taking a synbiotic for 8 weeks. Inclusion criteria were as follows: age over 18 years, chronic hemodialysis treatment, and ability to obtain adequate informed consent. Failure to obtain informed consent, having an active inflammatory disease, as well as intake of medications affecting the intestinal microbiome were regarded as exclusion criteria. Patients who had undergone antibiotic or chemotherapy in the previous three months, or were treated with biological agents during the same period, as well as taking phosphate binders, statins, and proton pump inhibitors were excluded from study.

Hemodialysis procedures were performed in line with the standard protocol using polyethersulfone high-flux dialyzer for each dialysis session. The applied synbiotic consists of 75 mg *Lactobacillus acidophilus* La-14 2 × 10^11^ CFU/g and 65 mg prebiotic fructooligosaccharides, taken once a day, 1 h after a meal. Patients were instructed not to alter their dietary habits, physical activities, or medication regimens. At the beginning of the study and again 8 weeks later, 5 mL of blood was drawn from each patient after an 8 h fasting. Blood samples were allowed to clot at room temperature and then centrifuged at 1000× *g* for 20 min. The serum was separated into Eppendorf microtubes and frozen at −80 °C for subsequent use.

### 2.2. Laboratory Analysis

Determination of serum concentrations of IS, para-CS, IL-6, and MDA was performed by enzyme-linked immunosorbent assay (ELISA). The assay was performed using commercial kits according to the manufacturer’s protocol (ELK Biotechnology, PRC; Denver, CO, USA).

### 2.3. Statistical Analysis

Statistical analyzes were performed using Statistical Package for Social Sciences (SPSS) Ver. 23 (SPSS, Inc., Chicago, IL, USA). All quantitative parameters were analyzed using Kolmogorov–Smirnov test. Unpaired and paired *t*-test were used to compare parameters between and within groups, respectively. Results were expressed as mean SD and differences were considered significant at *p* < 0.05.

## 3. Results

A total of 30 patients, 17 males and 13 females, were included in the study. No statistically significant difference was found based on sex (*p* = 0.638). The mean age of the subjects was 62.6 ± 14.3 years, again without any significant difference between men (60.4 ± 12.3 years) and women (63.1 ± 15.5 years) (*p* = 0.743).

In the Table 1 is illustrated the distribution of the studied population based on the frequency the dialysis procedures per week. The majority of patients on renal replacement therapy performed dialysis treatment three times a week with a duration of 4 h *(n* = 21). Due to preserved residual renal function, five of the patients are dialyzed twice a week for 4 h. The remaining four of patients were on a hemodialysis regimen three times a week with a duration of 3 h and 30 min.

Regarding the reasons for developing ESRD, the number of patients with chronic glomerulonephritis is the largest. Next are the patients with diabetic nephropathy and those with hypertensive nephroangiosclerosis (Table 2). As comorbidities, apart from secondary anemia and hyperparathyreoidism, more than half of the patients (*n* = 17) have only arterial hypertension. One third of them (*n* = 10) have diabetes and hypertension and three of the patients have hypertension and coronary heart disease as concomitant diseases. Nine patients had history of a cerebrovascular or cardiovascular event. The largest part of the patients (*n* = 14) have been undergoing hemodialisys treatment for more than 3 years, a little over 1/3 (*n* = 11) have been included in a dialysis program within the last year for at least 3 months, and the rest of the patients (*n* = 5) received renal replacement therapy for 1 to 3 years.

In the Table 3 and Table 4 are presented descriptive statistics of the studied indicators, before and after taking a synbiotic, in patients undergoing hemodialysis treatment. The statistical significance of the obtained results is also reflected. The serum levels of the two uremic toxins—IS and p-CS (Table 3 and Figure 1 and Figure 2)—decreased significantly after 8 weeks of synbiotic intake (*p* < 0.001 and *p* = 0.041). Similar results were established in the serum concentrations of MDA and IL-6 (Table 4 and Figure 3 and Figure 4) at the end of week 8 compared to the baseline levels (*p* < 0.001 and *p* < 0.001).

Table 5 represent the correlation between the evaluated uremic toxins prior and post-treatment and the frequency and duration of dialysis procedure.

Table 6 shows the values of certain blood parameters that we examine routinely in these patients. After two months of synbiotic supplementation, we found a significant difference (but within the normal range) in serum levels of total protein, potassium, and leukocyte count.

No adverse events were reported. All enrolled patients completed the study and were subjected to analysis.

## 4. Discussion

The current paper outlines a strong positive correlation between the intake of pro-, pre-, and synbiotics, and the metabolism of cresols and indoles. Such results are largely in line with the existing evidence on the matter [40,41]. In their systematic review, Nguyen TTU et al. [41] summarize a number of studies that demonstrate the positive impact of the gut microbiota modulators on the serum levels of p-cresyl sulfate in hemodialysis patients. They noted the insufficient number of and the need for further studies related to tracking serum levels of indoxyl sulfate in hemodialysis patients after probiotic or synbiotic supplementation. A significant association between dysbiotic gut microbiota and low-grade systemic inflammation in end-stage CKD patients has been demonstrated [42]. The pro-inflammatory biomarker IL-6, which was significantly decreased after taking a synbiotic in our study, appears to be the best predictor of cardiovascular risk and all-cause mortality in CKD [43]. Rossi et al. [44] found a correlation between increased values of IS and p-CS and increased pro-inflammatory biomarkers such as interleukin-6 and glutathione peroxidase in patients with CKD.

The patients on hemodialysis are subjected to enhanced oxidative stress as a result of elevated pro-oxidant activity [45] and inefficient antioxidant systems [46] related to the end stage of the renal disease as well as with the techniques of the hemodialysis treatment [47,48]. It is considered that the malondialdehyde generated from lipid peroxidation is a biomarker for increased oxidative stress in CKD [49], and its serum concentration in CKD patients is higher than that of healthy individuals [50,51]. Several studies have reported a potential antioxidant effect and decrease in oxidative stress biomarkers following therapeutic modulation of the gut microbiota. The systematic review by Nguyen TTU et al. [41] is also the first summary analysis of all studies related to monitoring MDA values in hemodialysis patients to date. In six of them, the beneficial effect of supplementation with probiotic or synbiotic on the serum levels of this oxidative stress marker was confirmed. The same effect was observed on serum concentrations of some of the inflammatory markers, including IL-6. The presented results are largely similar to the ones obtained in our study and support the conclusion derived by our research that the modulation of intestinal microbiota might have a positive effect on the levels of inflammatory biomarkers in patients with ESRD.

Despite the relatively small number of patients in our study, we analyzed the results obtained in the three subgroups depending on the frequency and the length of hemodialysis procedures (Table 5). It is noteworthy that the baseline values of protein-bound uremic toxins were lower in patients undergoing hemodialysis treatment three times a week for 4 h compared to the other two subgroups. In addition, achieving a significant reduction in IS and p-CS after synbiotic supplementation in patients with a lower weekly dialysis dose was more difficult than in patients undergoing hemodialysis treatment with a duration of 12 h per week. A similar finding was also observed when evaluating the serum levels of IL-6 and MDA. These results suggest that, although low, the clearance of protein-bound uremic toxins across the dialysis membrane is still of essence and the higher weekly dialysis dose leads to lower levels of their serum concentrations.

We also evaluated the values of IS and p-CS depending on the history of cerebrovascular or cardiovascular events in the study population. In patients with stroke or heart attack (*n* = 9), the baseline values of IS (248.87 ± 316.97 ng/mL) were higher compared to patients without such diseases (*n* = 21, 97.47 ± 148.55 ng/mL). We did not find the same correlation regarding baseline serum p-CS in patients with heart attack or stroke (13.03 ± 4.40 pg/mL) versus those without a cerebrovascular or cardiovascular event (36.22 ± 68.96 pg/mL).

In addition to the above-mentioned results related to our study, the tendency towards a decrease in serum creatinine levels (743.5 ± 224.11; 720.6 ± 214.45 µmol/L; *p* = 0.36) and urea (22.97 ± 5.55; 21.5 ± 3.51 mmol/L; *p* = 0.068) after synbiotic supplementation should be noted, although they did not reach significant values (Table 6).

It should be noted, though, that not all authors confirm the positive effect of the therapeutic modulation of the intestinal microbiota on the serum concentrations of biomarkers of inflammation and oxidative stress. For example, Hatakka et al. [52] did not find a significant decrease in CRP after taking a probiotic in patients with rheumatoid arthritis, and Lamprecht et al. [53] also did not find any effect of probiotic administration on serum MDA and IL-6 levels in trained men. The presence of such results is probably related to differences in the probiotic strains used, the applied probiotic dose, and the specificity of the dosage form taken.

The main disadvantages of the current study are the small population size and the lack of a control group to outline the causal relation between the medication intake and the decrease of toxin levels. The supplementation time probably also could be regarded as suboptimal. The parameters were measured only twice—at the beginning of the study and after 8 weeks. On the positive side we should outline the prospective nature of the research and the fact that there were no patients lost for follow-up, which undeniably increases the credibility of the results.

## 5. Conclusions

Synbiotic supplementation in patients with CKD undergoing conventional hemodialysis treatment leads to a decrease in the plasma levels of IS and p-CS. This has a beneficial effect on some of the complications of CKD, such as low-grade chronic inflammation and oxidative stress with a decrease in the serum concentrations of IL-6 and MDA. A high weekly dialysis dose of 12 h maintains lower serum levels of gut-delivered protein-bound toxins in hemodialysis patients despite their low clearance across the dialysis membrane. Additional multicenter studies are needed to specify the type, composition, and dose of the supplement used in patients with CKD. Although there is still no consensus regarding the duration and regimen of their intake, synbiotic supplementation can be used as an additional option in the treatment of hemodialysis patients.

## Figures and Tables

**Figure 1 medicina-59-01383-f001:**
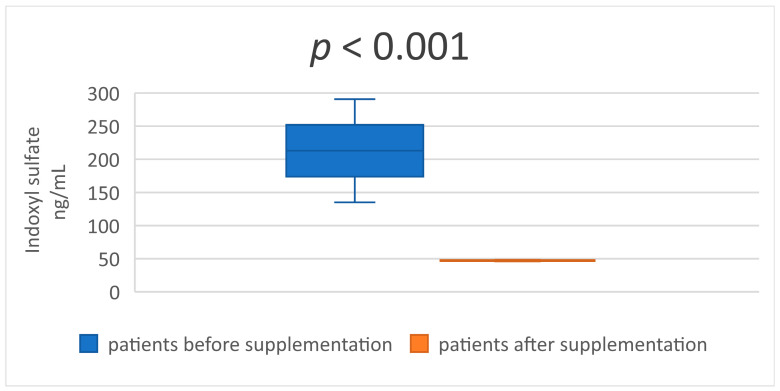
Mean value and confidence interval (CI) of serum indoxyl sulfate before and after supplementation.

**Figure 2 medicina-59-01383-f002:**
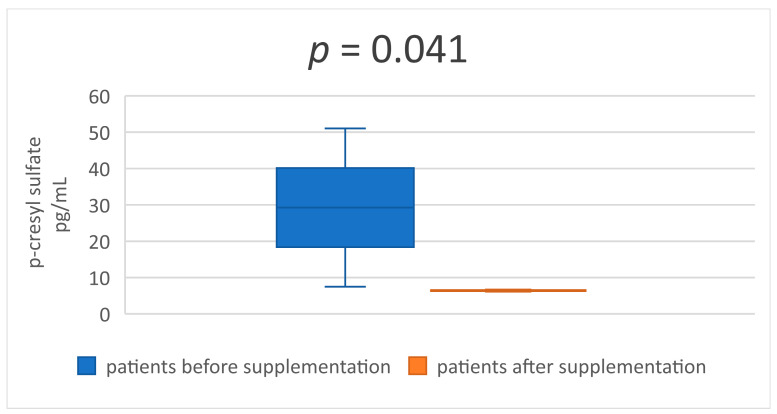
Mean value and CI of serum p-sresyl sulfate before and after supplementation.

**Figure 3 medicina-59-01383-f003:**
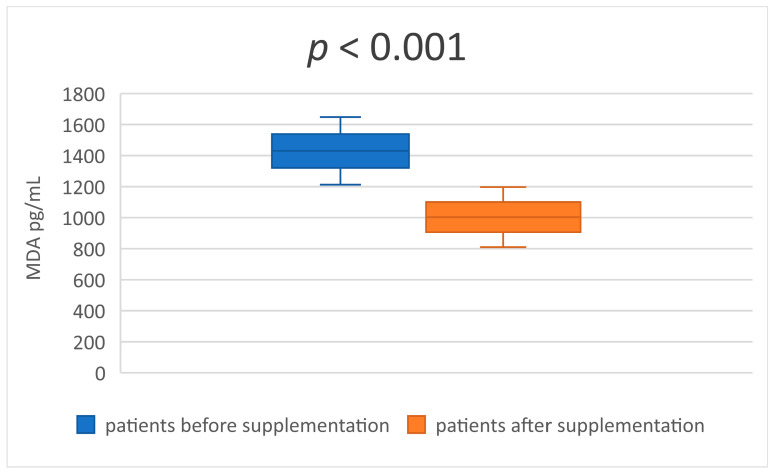
Mean value and CI of serum MDA before and after supplementation.

**Figure 4 medicina-59-01383-f004:**
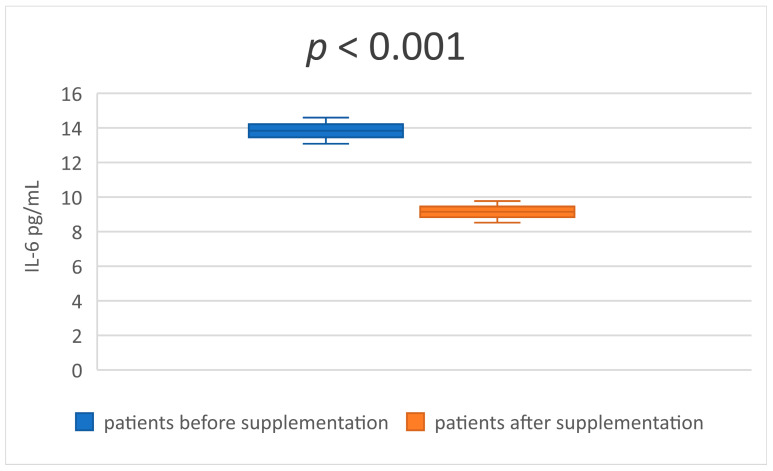
Mean value and CI of serum IL-6 before and after supplementation.

**Table 1 medicina-59-01383-t001:** Distribution of patients undergoing hemodialysis treatment according to the frequency of therapy.

Weekly Frequency and Duration of the Hemodialysis Procedure	Number of Patients
3 × 4 h	21
3 × 3 h and 30 min	4
2 × 4 h	5

**Table 2 medicina-59-01383-t002:** Distribution of patients undergoing hemodialysis treatment according to etiology of ESRD.

Kidney Disease	Number of Patients
Chronic glomerulonephritis	13
Diabetic nephropathy	6
Hypertensive nephroangiosclerosis	6
Autosomal dominant polycystic disease	3
Other	2

**Table 3 medicina-59-01383-t003:** IS and p-CS levels at baseline and at week 8.

	Indoxyl Sulfate ng/mL (Baseline)	Indoxyl Sulfate ng/mL (Week 8)	p-Cresyl Sulfate pg/mL (Baseline)	p-Cresyl Sulfate pg/mL (Week 8)
mean ± SD	212.89 ± 208.59	47.08 ± 3.24	29.26 ± 58.32	6.40 ± 0.79
95% confidence intervals	135.00–290.78	45.87–48.30	7.48–51.04	6.10–6.70
min.	42.48	40.94	6.17	6.17
max.	1012.28	54.14	265.00	8.43
SEM	38.08	0.59	10.64	0.14
*p*-value	*p* < 0.001 (paired-samples *t*-test)		*p* = 0.041 (paired-samples *t*-test)	

SEM—standard error of mean.

**Table 4 medicina-59-01383-t004:** IL-6 and Malondialdehyde (MDA) levels at baseline and at week 8.

	IL-6 pg/mL Before Taking a Synbiotic	IL-6 pg/mL after Intake	MDA pg/mL before Intake	MDA pg/mL after Intake
mean ± SD	13.84 ± 2.02	9.14 ± 1.67	1430.33 ± 583.42	1003.47 ± 518.37
95% confidence intervals	13.08–14.59	8.52–9.77	1212.48–1648.19	809.91–1197.04
min.	9.32	6.19	7.07	114.58
max.	22.46	11.72	2227.85	2046.74
SEM	0.37	0.30	106.51	94.64
*p*-value	*p* < 0.001 (paired-samples *t*-test)		*p* < 0.001 (paired-samples *t*-test)	

MDA—malondialedhyde, IL-6—interleukin-6.

**Table 5 medicina-59-01383-t005:** Values of IS, p-CS, IL-6, and MDA in the three subgroups of hemodialysis patients according to the frequency and length of the dialysis treatment.

Weekly Frequency and Duration of HD Procedure	Parameters Before and After Synbiotic Intake	*n*	Mean ± SD	*p*-Value *
3 × 4 h	IS ng/mL before	21	159.88 ± 150.925	0.003
	IS ng/mL after		47.56 ± 3.38	
	p-CS pg/ml before	21	14.3 ± 7.74	0.001
	p-CS pg/mL after		6.25 ± 0.66	
	IL-6 pg/mL before	21	13.65 ± 1.26	0.001
	IL-6 pg/mL after		9.16 ± 1.78	
	MDA pg/mL before	21	1307.09 ± 617.18	0.001
	MDA pg/mL after		422.37 ± 592.07	
3 × 3 h and 30 min	IS ng/mL before	4	407.19 ± 406.44	0.176
	IS ng/mL after		47.40 ± 2.41	
	p-CS pg/ml before	4	16.64 ±7.14	0.058
	p-CS pg/mL after		6.28 ± 0.46	
	IL-6 pg/mL before	4	15.05 ± 5.09	0.109
	IL-6 pg/mL after		8.82 ± 1.51	
	MDA pg/mL before	4	1657.88 ± 529.77	0.012
	MDA pg/mL after		163.42 ± 326.85	
2 × 4 h	IS ng/mL before	5	280.10 ± 139.68	0.019
	IS ng/mL after		44.84 ± 2.72	
	p-CS pg/ml before	5	102.23 ± 127.81	0.173
	p-CS pg/mL after		7.14 ± 1.19	
	IL-6 pg/mL before	5	13.66 ± 0.80	0.002
	IL-6 pg/mL after		9.32 ± 1.57	
	MDA pg/mL before	5	1765.90 ± 283.70	0.191
	MDA pg/mL after		1047.47 ± 976.30	

*—paired-samples *t*-test.

**Table 6 medicina-59-01383-t006:** Serum levels of some blood parameters before and after synbiotic intake in hemodialysis patients.

Blood Parameters	Mean	*n*	Std. Deviation	Std. Error Mean	*p*-Value *
HGB g/L before intake HGB g/L after intake	102.56	30	15.35	2.80	0.293
	100.30		17.46	3.18	
RBC 10^12^/L before intake RBC 10^12^/L after intake	3.45	30	0.59	0.10	0.142
	3.33		0.69	0.10	
WBC 10^9^/L before intake WBC 10^9^/L after intake	8.54	30	3.17	0.57	0.05
	7.62		2.07	0.37	
PLT 10^9^/L before intake PLT 10^9^/L after intake	218.9	30	120.17	21.9	0.711
	214.9		103.2	18.84	
Urea mmol/L before intake Urea mmol/L after intake	22.97	30	5.55	1.01	0.068
	21.5		3.51	0.64	
Creatinine µmol/L before intake Creatinine µmol/L after intake	743.5	30	224.11	40.91	0.136
	720.6		214.45	39.15	
T.protein g/L before intake T.protein g/L after intake	68.33	30	6.77	1.23	0.025
	66.46		5.8	1.06	
K mmol/L before intake K mmol/L after intake	5.45	30	0.85	0.15	0.031
	5.2		0.88	0.16	
T.Ca mmol/L before intake T.Ca mmol/L after intake	2.31	30	0.22	0.04	0.239
	2.29		0.23	0.04	
P mmol/L before intake P mmol/L after intake	1.86	30	0.54	0.09	0.339
	1.96		0.53	0.09	
Fe µmol/L before intake Fe µmol/L after intake	8.54	30	2.91	0.53	0.294
	9.12		2.8	0.51	

*—paired-samples *t*-test.

## Data Availability

The data presented in this study are available upon request from the corresponding author. The data are not publicly available due to national legal restrictions.

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
