# Peer review of "The Effect of Synbiotic Supplementation on Uremic Toxins, Oxidative Stress, and Inflammation in Hemodialysis Patients—Results of an Uncontrolled Prospective Single-Arm Study"

_medicina, 2023, doi:10.3390/medicina59081383_

Round 1

Reviewer 1 Report

Review of the manuscript medicina-2472679

The Effect of Synbiotic Supplementation on Uremic Toxins, Oxidative Stress and Inflammation in Hemodialysis Patients

By Kuskunov T. et al.

The reviewed manuscript is an original paper. The authors evaluate the effect of an 8-week synbiotic supplementation in hemodialysis patients with chronic kidney disease. As a result of the administration of the synbiotic, a reduction in markers of inflammation and oxidative stress (interleukin-6 and malondialdehyde; MDA) and bacterial metabolites: indoxyl sulfate (IS) and p-cresyl sulfate (p-CS) was observed in the blood of patients. This confirms the general beneficial effect of synbiotic supplementation on the course of CKD. Thus, the manuscript is another paper from numerous others confirming the concept of beneficial effects resulting from alleviating intestinal dysbiosis in CKD patients as a result of the use of probiotic/prebiotic/synbiotic preparations.

MAJOR REMARKS

1. Introduction – the authors describe the altered composition of the gastrointestinal microbiota. It is also worth describing the physiological composition (state of eubiosis) and present the state of dysbiosis against this background.

2. Introduction – line 82-83. Oxidative stress in CKD should be described in more detail and briefly the possibilities of its pharmacological correction based on publications, e.g..

Medical Studies/Studia Medyczne 2022; 38 (2): 163–170  DOI: https://doi.org/10.5114/ms.2022.117714

 Oxid Med Cell Longev. 2020; 2020: 5478708. doi: 10.1155/2020/5478708

 Antioxidants 2020, 9(8), 752; https://doi.org/10.3390/antiox9080752

3. Materials and methods – the study population should be characterized in more detail. In terms of comorbidities, reasons for developing CKD and the need for hemodialysis, medications used, etc.

4. Material and methods – what drugs other than those listed above were the reason for exclusion from the study? What about drugs that increase oxidative stress?

5. Results – results should be reported as mean OR median. For normally distributed data, you can analyze the mean along with the standard deviation. Otherwise, the median. You must provide the results of the distribution analysis.

6. Discussion – The authors describe a number of other papers that have previously demonstrated the benefits of probiotic/synbiotic supplementation and reduced intestinal dysbiosis in CKD patients. In this context, the authors' paper is not innovative and does not bring anything new. It would therefore be worth presenting the results in a slightly different way - for example, by correlating laboratory results with the frequency of dialysis or after different lengths of supplementation. Then the manuscript would gain from novelty.

7. The final part of the discussion - other aspects that indicate the weaknesses of the reported study should also be indicated - small population size, short period of supplementation etc.

MINOR REMARKS

1. Abstract and text of the manuscript - the expression describing the number of bacteria in synbiotic preparation should be given in exponential notation, not as the word "billion"

2. Results – the pharmaceutical form of the synbiotic used should be described in more detail

3. The list of references dose not follow the Journal's guidelines (numerous errors in the bibliographic description of the cited journals)

4. Tables have not been prepared in accordance with the guidelines (e.g. vertical division lines)

Author Response

Response to Reviewer 1 Comments

Thank you very much for reviewing our manuscript. I find your remarks to be well-founded and largely justified and increasing the scientific value of the article.

Point 1: Introduction – the authors describe the altered composition of the gastrointestinal microbiota. It is also worth describing the physiological composition (state of eubiosis) and present the state of dysbiosis against this background.

Response 1: A brief description of normal intestinal microbiota was provided in the revised version.

Point 2: Introduction – line 82-83. Oxidative stress in CKD should be described in more detail and briefly the possibilities of its pharmacological correction based on publications, e.g..

Response 2: Upon review certain issues were raised by Reviewer, suggesting that the Introduction section should be more consice. This fact precluded significant expansion of the Introduction section. We hope that it doesn’t impair the value of the article in general.

Point 3: Materials and methods – the study population should be characterized in more detail. In terms of comorbidities, reasons for developing CKD and the need for hemodialysis, medications used, etc.

Response 3: More detailed description of the studied population was provided in the Results section

Point 4: Material and methods – what drugs other than those listed above were the reason for exclusion from the study? What about drugs that increase oxidative stress?

Response 4: Information provided as required.

Point 5: Results – results should be reported as mean OR median. For normally distributed data, you can analyze the mean along with the standard deviation. Otherwise, the median. You must provide the results of the distribution analysis.

Response 5: Additional information provided in the revised version.

Point 6: Discussion – The authors describe a number of other papers that have previously demonstrated the benefits of probiotic/synbiotic supplementation and reduced intestinal dysbiosis in CKD patients. In this context, the authors' paper is not innovative and does not bring anything new. It would therefore be worth presenting the results in a slightly different way - for example, by correlating laboratory results with the frequency of dialysis or after different lengths of supplementation. Then the manuscript would gain from novelty.

Response 6: Unfortunately the sample size wasn’t sufficient to derive statistically significant correlation with dyalisis frequency, while the study design included only one group of patients based on duration of supplementation. Our intent is to assess the influence of those factors in subsequent research on the matter.

Point 6: Minor remarks

Response 6: We applied corrections where recommended. Should further suggestions occur they’ll be reflected in the subsequent revisions.

Reviewer 2 Report

In the manuscript “The effect of synbiotic supplementation on uremic toxins, oxidative stress and inflammation in hemodialysis patients” the authors present the results of the study on the effect of symbiotic/prebitoic on the serum levels of indoxyl sulfate (IS) and p-cresyl sulfate (p-CS) as well as on serum levels of  biomarkers of inflammation and oxidative stress  in hemodialysis patients. There are a number of studies on the effects of probiotics, prebiotics, and synbiotics on uremic toxins, inflammation, and oxidative stress in hemodialysis patients as well as  recently published a systematic review and meta-analysis of randomized controlled trials (Nguyen TTU,  et al. doi: 10.3390/jcm10194456; not cited in the manuscript). Therefore this is not an original study although the results presented are consistent with those previously published and confirm the beneficial effects of the use of synbiotics in hemodialysis patients.

I also have a few additional comments.

1.     The introduction in the abstract is too long and the aim of the study is not written.

2.     The introduction  in the text is also too extensive and should be condensed, and certain parts of the existing introduction could be moved to the discussion.

3.     The hemodialysis procedure should be described in more detail.

4.     Instead of Figure 1, it would be better to present a table with main characteristics of patients, including demographic data, HD vintage, number of HD and hours of HD per week, and basic laboratory data (blood count, serum level of creatinine , urea, etc.)

5.     A large number of references were cited, and almost all cited articles were published more than five years ago. It would be better if some earlier references were replaced with new ones published in the last few years.

6.     Abbreviations should be defined the first time they appear in the text although they are explained in the abstract.

Author Response

Response to Reviewer 2 Comments

Thank you very much for reviewing our manuscript. I find your remarks to be well-founded and largely justified and increasing the scientific value of the article.

Point 1: The introduction in the abstract is too long and the aim of the study is not written.

Response 1: The aim of the study was added in the abstract.

Point 2: The introduction in the text is also too extensive and should be condensed, and certain parts of the existing introduction could be moved to the discussion.

Response 2: Upon review certain issues were raised by Reviewer, suggesting additional details in the Introduction section. This fact largley precluded significant reduction of the Introduction section. We hope that it doesn’t impair the value of the article in general.

Point 3:  The hemodialysis procedure should be described in more detail.

Response 3: Additional information on the dialysis regimen of the studied population was provided.

Point 4: Instead of Figure 1, it would be better to present a table with main characteristics of patients, including demographic data, HD vintage, number of HD and hours of HD per week, and basic laboratory data (blood count, serum level of creatinine , urea, etc.)

Response 4: Information included in the revised version as recommended.

Point 5: A large number of references were cited, and almost all cited articles were published more than five years ago. It would be better if some earlier references were replaced with new ones published in the last few years.

Response 5: Additional references were included.

Point 6: Abbreviations should be defined the first time they appear in the text although they are explained in the abstract.

Response 6: Remark noted and correction provided in the revised version.

Round 2

Reviewer 1 Report

Re-review of the manuscript

medicina-2472679-peer-review-v2

THE EFFECT OF SYNBIOTIC SUPPLEMENTATION ON UREMIC TOXINS, OXIDATIVE STRESS AND INFLAMMATION IN HEMODIALYSIS PATIENTS

By Teodor Kuskunov et al.

Thank you for submitting the revised version of the manuscript.

The authors largely responded to my comments in the previous review.

However, my reservation concerning the novelty of the results presented in the manuscript is still valid. The authors describe their results confirming the thesis that has been proven many times in many other works, mention in the discussion by the authors themselves. Therefore, I think it would be reasonable to try to demonstrate the obtained results in a slightly different light - perhaps at least presenting the results of the evaluated compounds (IS and p-CS ) in 3 subgroups distinguished on the basis of frequency and length of dialysis? With the reservation that these are some kind of preliminary analysis due to the significantly limited number of patients in these subgroups?

Other remarks:

1. The new fragment introduced into the chapter "Results" (lines 173-191) significantly differs in term of English language quality from the rest of the manuscript. For some unknown reason, this fragment has been written in the present tense. The quality and style of this passage needs urgent improvement.

2. Figures 1-4 showing changes in IS, p-CS, MDA and Il-6 before and after supplementation in their current form are unacceptable. They look more like screenshots than professionally imported charts into text. The quality (resolution) is very poor.

Moreover, Authors should also include the result of the statistical analysis (p-value) in the caption to each figure, and mark the statistical significance in the figure itself with, for example, an asterisk. Authors should also describe in the caption what the graphs show - mean +/- SD ???? (what are "whiskers" on the graphs???)

Author Response

Response to Reviewer 1

Point 1. 1. The new fragment introduced into the chapter "Results" (lines 173-191) significantly differs in term of English language quality from the rest of the manuscript. For some unknown reason, this fragment has been written in the present tense. The quality and style of this passage needs urgent improvement.

Response 1: The language was revised and corrected in line with the recommendations.

Point 2: Figures 1-4 showing changes in IS, p-CS, MDA and Il-6 before and after supplementation in their current form are unacceptable.They look more like screenshots than professionally imported charts into text.The quality (resolution) is very poor.

Response 2: An improved and more concise version of the figures was provided in the revised version.

Point 3: Moreover, Authors should also include the result of the statistical analysis (p-value) in the caption to each figure, and mark the statistical significance in the figure itself with, for example, an asterisk. Authors should also describe in the caption what the graphs show - mean +/- SD ???? (what are "whiskers" on the graphs???)

Response 3: Adjustments applied in the revision.